# P2X7 Receptors in Astrocytes: A Switch for Ischemic Tolerance

**DOI:** 10.3390/molecules27123655

**Published:** 2022-06-07

**Authors:** Yuri Hirayama, Naohiko Anzai, Hiroyuki Kinouchi, Schuichi Koizumi

**Affiliations:** 1Department of Pharmacology, Chiba University Graduate School of Medicine, 1-8-1, Inohana, Chuo-ku, Chiba-shi 260-8670, Chiba, Japan; yhirayama@chiba-u.jp (Y.H.); afga5078@faculty.gs.chiba-u.jp (N.A.); 2Department of Neuropharmacology, Interdisciplinary Graduate School of Medicine, University of Yamanashi, 1110 Shimokato, Chuo 409-3898, Yamanashi, Japan; 3Department of Neurosurgery, Interdisciplinary Graduate School of Medicine, University of Yamanashi, 1110 Shimokato, Chuo 409-3898, Yamanashi, Japan; hkinouchi@yamanashi.ac.jp; 4Yamanashi GLIA Center, University of Yamanashi, 1110 Shimokato, Chuo 409-3898, Yamanashi, Japan

**Keywords:** P2X7 receptor, ischemic tolerance, astrocytes

## Abstract

A sub-lethal ischemic episode (preconditioning [PC]) protects neurons against a subsequent lethal ischemic injury. This phenomenon is known as ischemic tolerance. PC itself does not cause brain damage, but affects glial responses, especially astrocytes, and transforms them into an ischemia-resistant phenotype. P2X7 receptors (P2X7Rs) in astrocytes play essential roles in PC. Although P2X7Rs trigger inflammatory and toxic responses, PC-induced P2X7Rs in astrocytes function as a switch to protect the brain against ischemia. In this review, we focus on P2X7Rs and summarize recent developments on how astrocytes control P2X7Rs and what molecular mechanisms they use to induce ischemic tolerance.

## 1. Introduction

The brain is one of the most vulnerable organs to ischemia. Therefore, scientists have been pursuing research to save the brain against ischemia, and have also spent a great deal of time and money developing drugs to treat stroke. There have been more than 1000 clinical trials on stroke targeting neurons, but most of them have failed [1]. Dr. Barres believes that a neuron-related strategy is insufficient to save the brain and will not result in effective therapeutic drugs for stroke. He has stated “Glial cells know how to save the brain, but researchers have not known yet” [2]. Despite the difficulties encountered in developing drugs and therapies for stroke, major progress has been made in research on ischemic tolerance. In this phenomenon, organs that experience prior mild non-invasive ischemic preconditioning (PC) acquire tolerance to subsequent invasive ischemic stress. This ischemic tolerance is commonly observed clinically and experimentally. The endogenous neuroprotective effects by PC were originally reported in the heart [3,4], but were also observed in the kidneys [5], liver [6], skeletal muscle [7], and the brain [8,9]. Since the discovery of ischemic tolerance [3], it has received tremendous attention because it shows robust neuroprotective effects. With regard to cerebral ischemic tolerance, there have been a large number of studies about mechanisms of ischemic tolerance [10,11], but almost all studies were performed from the point of view of neurons. Recently, it has been revealed that glial cells play crucial roles in regulating central nervous system (CNS) injury and recovery including cerebral ischemic tolerance [12,13,14], but the molecular mechanisms of cerebral ischemic tolerance are poorly understood. Our recent studies have shown that glial cells, especially astrocytes, play an essential role in the induction of ischemic tolerance [15,16]. Extracellular ATP (eATP) and astrocytic P2X7 receptors (P2X7Rs), which are ATP-gated ion channels, play indispensable roles in the molecular mechanisms of astrocyte-mediated ischemic tolerance. In this review, we summarize astrocyte-mediated ischemic tolerance and focus on P2X7Rs.

## 2. Localization and Functions of P2X7Rs

Purinergic signaling was proposed as extracellular signaling molecules in 1972, and recently focus has been put on the therapeutic potential of both P1 (adenosine) and P2 receptors [17]. For example, P2Y12 is a G protein-coupled receptor, and its antagonists inhibit aggregation in platelets and thus are widely used for the treatment of thrombosis and stroke [18]. Among seven subtypes of P2X ion channel receptors, P2X7Rs are a non-selective cation channel gated eATP, and it has been revealed that they play a crucial role in the CNS [19]. Although P2X7Rs are ion channels, they differ from other subtypes of P2X receptors in that P2X7Rs are much less sensitive to eATP, require ~mM eATP to be activated, have a long intracellular C terminus, and form a large pore when activated [20]. Therefore, the activation of P2X7Rs not only increases cation permeability, but also increases the permeability of larger molecules and various C-terminus-mediated intracellular signal cascades. These cascades include phosphatidylinositol 3-kinase/Akt, extracellular signal-regulated kinase, and mitogen-activated protein kinases [21,22]. Therefore, the roles of P2X7Rs are diverse and control various physiological and pathological events. These events include the release of proinflammatory cytokines, such as tumor necrosis factor [23] and interleukin-1β [24], proliferation [25], induction of cell death [26], phagocytosis [27], and inflammatory responses [28].

In the adult brain, P2X7Rs are mainly expressed in microglia. In physiological conditions, P2X7Rs are not active simply because eATP in the healthy brain is insufficient to activate these weakly sensitive P2 receptors [29,30]. Additionally, the findings that P2X7R knockout mice are healthy and have no major phenotype in physiological conditions [31] support the idea that P2X7Rs do not function well in the healthy brain. However, unlike physiological conditions, P2X7Rs are upregulated and activated in various pathological conditions or diseases (Table 1). P2X7Rs are associated with the pathological process of neuropathic pain via inflammatory responses, such as the release of tumor necrosis factor-α and interleukin-1β. In spared nerve injury, which is a neuropathic pain model, P2X7Rs are increased in microglia, and a P2X7R antagonist can suppress the development of mechanical hypersensitivity [32].

P2X7Rs are also upregulated in other types of cells, such as neurons. In status epileptics induced by kainic acid, P2X7Rs are increased in dentate granule neurons, and kainic acid-evoked status epileptics are inhibited by P2X7R antagonists, suggesting the causal role of neuronal P2X7Rs in epilepsy [40]. In multiple sclerosis, P2X7Rs are increased in oligodendrocytes, and a P2X7R antagonist can reduce demyelination by chronic experimental autoimmune encephalomyelitis, which is a model of the disorder [34]. In a Huntington’s disease mouse model, R6/1 mice, P2X7Rs are upregulated in neurons and microglia, and the administration of a P2X7R antagonist to R6/1 mice attenuates body weight loss and a motor coordination deficit [41]. Therefore, P2X7Rs appear to be associated with pathological events, and function as a “death receptor” or “toxic receptor”. Several clinical studies have been performed to test the efficacy of P2X7R inhibitors on pathological events [42].

However, P2X7Rs also have beneficial roles in some pathological brains. For example, it has been reported that the activation of P2X7R by ATP induced the release of tumor necrosis factor-α from microglia, which protected neurons from *N*-methyl-D-aspartate-induced excitotoxicity in organotypic hippocampal slice cultures [43]. In a cerebellar granule neuron culture, Ortega et al. showed that glutamate-induced cell death was prevented by P2X7R agonist BzATP [44]. They also showed that BzATP elicited the neuroprotection of granule neurons via a phosphorylation of GSK3-mediated mechanism(s) [45]. In CNS diseases, the activation of the pannexin 1/P2X7R complex contributes to the neuroprotective mechanism of ischemic postconditioning [33]. Similarly to this effect, our recent studies have shown that after mild, non-invasive brain ischemic PC, P2X7Rs are upregulated and have a central role in inducing “ischemic tolerance”. Interestingly, after PC, P2X7Rs are mainly upregulated in astrocytes [15]. Furthermore, although ischemic tolerance is believed to be caused by cell autonomous mechanisms of neurons, astrocytes play a main role in its induction. Additionally, P2X7Rs play a major role in regulating astrocyte-mediated ischemic tolerance. Therefore, P2X7Rs are not solely toxic or death receptors, but are a double-edged sword to control the pathological brain. In the following sections, we summarize the latest findings on astrocyte-dependent ischemic tolerance, focusing on the beneficial role of P2X7Rs.

## 3. Astrocytic P2X7R-Mediated Ischemic Tolerance

The middle cerebral artery occlusion model is widely used to induce a focal cerebral ischemia/reperfusion injury in mice [46]. Using this model, we found that P2X7Rs played a neuroprotective role in cerebral ischemic tolerance in astrocytes [15]. Noninvasive mild ischemia PC significantly reduced the subsequent severe ischemia-induced infarct area (i.e., induction of ischemic tolerance). The region of PC-induced activated astrocytes was well correlated with that of the reduced cerebral infarction by PC. Activated astrocytes, which are induced by ischemia, exhibit a protective phenotype [47]. Furthermore, several researchers have suggested the neuroprotective role of astrocytes in ischemic tolerance [48]. A previous study reported that chemical preconditioning with 3-nitropropionic acid induced tolerance to ischemia by activating astrocytes [49]. Therefore, we investigated whether PC-induced astrocytic activation was essential for ischemic tolerance and found that the inhibition of astrocytic activation by fluorocitrate [50] abolished ischemic tolerance. This finding suggested that PC transforms astrocytes into an ischemia-resistant phenotype and that they lead to ischemic tolerance.

With regard to the molecular mechanisms, we focused on ATP, which is the main molecule involved in neuron–glia interactions in CNS diseases, and examined the expression of the receptors important for ischemic tolerance. In cerebral ischemia, the postischemic upregulation of P2X (P2X2, 4, and 7) and P2Y (P2Y1) receptor subtypes on neuron and glial cells has been reported [51]. Among these receptors, we found that P2X7Rs in astrocytes were significantly upregulated by PC, and ischemic tolerance was abolished in P2X7R knockout mice, despite activating astrocytes. Notably, there was no significant difference in the vulnerability to severe ischemia between wild-type and P2X7R knockout mice, which is consistent with other reports [52,53]. To determine the molecular mechanisms involved in astrocytic P2X7R-mediated ischemic tolerance, we focused on hypoxia-inducible factor-1α (HIF-1α), which is a mediator of ischemic tolerance [54]. HIF-1α is a transcription factor that regulates the expression of various neuroprotective molecules, such as erythropoietin [55,56,57], and cells overexpressing P2X7R upregulate HIF-1α expression [58]. Therefore, we speculate that astrocytic P2X7Rs provide neuroprotection against severe ischemia via HIF-1α target molecules and induce ischemic tolerance. Predictably, studies showed that astrocytic HIF-1α was persistently upregulated after PC in a P2X7R-dependent manner and induced the expression of neuroprotective molecules including erythropoietin, thereby leading to ischemic tolerance [15,59]. Therefore, the P2X7R-mediated upregulation of HIF-1α in astrocytes is essential for PC-induced ischemic tolerance.

Although PC also increases P2X7Rs in microglia, the inhibition of PC-induced microglial activation and subsequent upregulation of P2X7Rs by minocycline [60] have no effect on ischemic tolerance. This finding suggests that microglial P2X7Rs are not involved in ischemic tolerance [15]. However, the reason why astrocytic, but not microglial, P2X7Rs are important is unclear. Additionally, P2X7Rs require much higher concentrations of eATP for their activation, but the PC-evoked increase in eATP is insufficient to activate P2X7Rs. Therefore, although the importance of astrocytic P2X7Rs in ischemic tolerance has been clarified, the major questions remain (1) why microglial P2X7Rs are not involved and (2) why astrocytic P2X7Rs function when eATP is insufficient to activate P2X7Rs by PC. Recently, it has been found that these questions were solved by ecto-ADP-ribosyltransferase 2 (ARTC2) [61].

## 4. Mechanisms of P2X7R Activation

Recently, studies reported another mechanism of P2X7R activation that does not require high eATP concentrations [62,63]. In peripheral T cells that express ARTC2, P2X7Rs can be activated by low eATP concentrations if nicotinamide adenine dinucleotide (NAD^+^) is present. NAD^+^ is a substrate for ARTC2 and is required to activate P2X7Rs via ARTC2-catalyzed ADP-ribosylation. Therefore, NAD^+^ increases the sensitivity of P2X7Rs to eATP in ARTC2-expressing cells. Additionally, unlike ATP, NAD^+^ covalently modifies P2X7Rs, which results in the persistent activation of P2X7Rs via NAD^+^/ARTC2 signaling [64]. Although the characteristics of ARTC2 in peripheral immune cells, such as T cells, have been well researched, the localization and function of ARTC2 in the brain remain unknown.

In our recent study, we unraveled the crucial role of ARTC2 in cerebral ischemic tolerance [61]. Although ARTC2-positive signals were not observed in the naïve brain, they were significantly increased in astrocytes by PC. In in vitro experiments using primary cultures of astrocytes that expressed ARTC2, NAD^+^ increased the sensitivity of P2X7Rs to eATP. This finding suggests that ARTC2 acts as a catalyst for ADP-ribosylation in astrocytes. Furthermore, the PC-evoked activation of P2X7Rs and a subsequent increase in HIF-1α in astrocytes were abolished by the ARTC2 inhibitor S+16a [65]. This inhibitor also suppressed the induction of ischemic tolerance. Therefore, these findings indicate that astrocytes sensitize P2X7Rs to eATP via an NAD^+^/ARTC2-mediated mechanism after PC. Additionally, astrocytes induce ischemic tolerance, even when the eATP concentration is not sufficient to activate P2X7Rs under PC-evoked conditions. Notably, microglia do not possess this mechanism because ARTC2 is absent in microglia. Kahl et al. (2000) reported that ARTC2 in T cells is removed by a metalloproteinase after activation [66]. This finding raises the possibility that, similar to T cells, microglial ARTC2 is removed after PC. Therefore, astrocytic, but not microglial, P2X7Rs are activated after PC, and play an important role in ischemic tolerance.

We also showed that P2X7R-mediated responses were significantly increased by NAD^+^ in astrocyte culture [61]. However, Rissiek et al. (2020) reported that the dominant P2X7R in mouse astrocytes in primary culture was P2X7a, which is a splice variant that is much less sensitive to NAD^+^/ARTC2-mediated ADP-ribosylation [67]. They also showed that ADP-ribosylation triggered the gating of P2X7k, which is another splice variant, but not P2X7a [68]. We speculate that these discrepancies between the two groups could be due to the experimental conditions. We reported that P2X7Rs were rarely expressed in normal astrocytes in vivo but were expressed under in vitro conditions, such as primary culture [15]. Therefore, differences in the extracellular environment, e.g., culture conditions, may affect the splicing of P2X7R, producing distinct splice variants with different sensitivities to ADP-ribosylation. In fact, we used a single cell culture [61], whereas Rissiek et al. [67] used mixed glial cell cultures. Additionally, splice variants of P2X7 vary greatly among mouse strains [69]. These factors might also affect the sensitivity to NAD^+^ and explain the discrepancies between Rissiek et al.’s study [67] and our study. Therefore, the variants of P2X7Rs are diverse and their regulation is complex. Further studies are necessary to clarify the molecular mechanisms of P2X7R activation via NAD^+^/ARTC2 signaling.

## 5. Conclusions

As shown in Figure 1, this article describes the latest findings on the protective role of astrocytic P2X7Rs in cerebral ischemic tolerance. In pathological conditions, P2X7Rs expressed in microglia and neurons are activated by much higher concentrations of eATP than those under normal conditions. Additionally, P2X7Rs function as a death receptor via inflammatory and toxic responses. However, in cerebral ischemic tolerance, astrocytes can sensitize P2X7Rs to eATP via ARTC2-catalyzed ADP-ribosylation. Therefore, astrocytic P2X7Rs are activated by low concentrations of eATP, such as under PC-evoked conditions, and function as a protective receptor against ischemic injury. These findings suggest that the manner of activation and subsequent downstream events of P2X7Rs depend on the expressing cell types. Therefore, we conclude that P2X7Rs have a dual nature and that astrocytic P2X7Rs are expected to be a promising therapeutic target for preventing cerebral ischemic injury.

Astrocytes and microglia were activated by preconditioning (PC) and thus upregulated P2X7 receptor (P2X7R) levels. Under normal conditions, the P2X7R requires a high extracellular ATP (eATP) concentration for activation. Although eATP levels after PC were not sufficient to activate P2X7Rs, astrocytes could sensitize P2X7Rs to eATP by nicotinamide adenine dinucleotide (NAD^+^)/ecto-ADP-ribosyltransferase 2 (ARTC2)-mediated modification. The ADP-ribosylated P2X7R can be activated by low eATP levels released by non-invasive stimuli, such as PC, and the result can be sustained. This allows for the long-lasting upregulation of hypoxia inducible factor-1α (HIF-1α) in astrocytes and the induction of ischemic tolerance. The presence or absence of ARTC2 may also define the different roles of astrocytes and microglia.

## Figures and Tables

**Figure 1 molecules-27-03655-f001:**
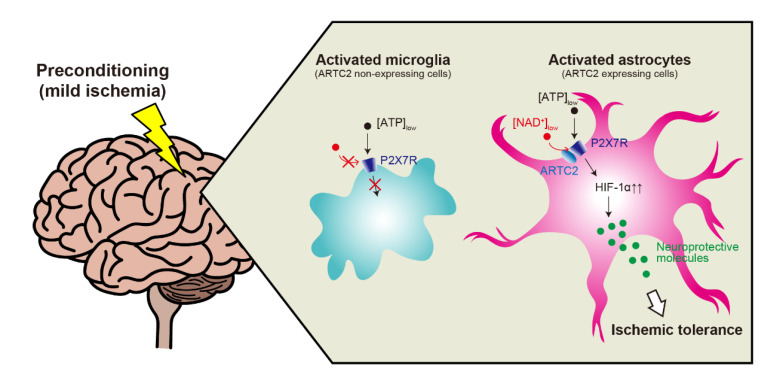
Schematic diagram of the mechanisms underlying P2X7 receptor activation in astrocytes and microglia.

**Table 1 molecules-27-03655-t001:** Role of P2X7 receptors in central nervous system diseases.

Roles	Pathology (In Vivo Model)	Findings	Ref.
Protective	Cerebral ischemic tolerance by preconditioning (MCAO)	Cerebral ischemic tolerance is abolished in P2X7R knock-out mice	[15]
	Cerebral ischemic tolerance by postconditioning (BCAO)	Ischemic postconditioning-induced neuroprotective effects are abolished by pretreatment of pannexin 1/P2X7R antagonist mefloquine	[33]
Harmful	Multiple sclerosis (EAE)	BBG or oATP ameliorates chronic EAE by reducing demyelination	[34]
	ALS (SOD1-G93A mice)	BBG attenuates motor neuron loss in SOD1-G93A mice	[35]
	Parkinson’s disease (6-OHDA)	BBG attenuates the 6-OHDA-induced neurotoxicity	[36]
	Alzheimer’s disease (hAPP-J20 mice)	BBG prevents the development of amyloid plaques in hAPP-J20 mice	[37]
	Neuropathic pain (SNI, PSL, and SNL)	P2X7R antagonist A-438079 suppresses the development of mechanical hypersensitivity in SNI model	[32]
		Development of both thermal and mechanical hypersensitivity after PSL is absent in P2X7R knock-out mice	[38]
		P2X7R antagonist A-740003 reduces SNL-induced mechanical allodynia	[39]
	Status epilepticus (KA)	BBG or P2X7R antagonist A438079 protects against KA-induced neuronal death	[40]
	Huntington’s disease (R6/1 mice)	Administration of BBG to R6/1 mice attenuates their motor-coordination deficit	[41]

Abbreviations: P2X7R, P2X7 receptor; MCAO, middle cerebral artery occlusion; BCAO, bilateral carotid artery occlusion; EAE, experimental autoimmune encephalomyelitis; BBG, brilliant blue G (P2X7R antagonist); oATP, oxidized ATP (P2X7R antagonist); ALS, amyotrophic lateral sclerosis; 6-OHDA, 6-hydroxydopamine; SNI, spared nerve injury; PSL, partial sciatic nerve ligation; SNL, spinal nerve ligation; KA, kainic acid.

## Data Availability

The data presented in this study are available on request from the corresponding author.

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
