# Peer review of "P2X7 Receptors in Astrocytes: A Switch for Ischemic Tolerance"

_molecules, 2022, doi:10.3390/molecules27123655_

Round 1

Reviewer 1 Report

The paper according to the title appeared to be interesting. However, after its reading and revision, I consider that it requires an extensive revision and reorganization. Some statements/sentences along the manuscript do not agree with data reported in the literature. For example, in the Introduction section “Although glial cells have crucial roles in regulating central nervous system (CNS) injury and recovery, their contribution to cerebral ischemic tolerance remains largely unknown”. This is not true. There is a great body of evidences showing the involvement of glial cells in brain ischemia and hypoxic events as well as the implication of different nucleotide receptors. Moreover, authors have recently reported a review article summarizing the roles of astrocytes in brain ischemia (Koizumi S, Hirayama Y and Morizawa YM, 2018, New roles of reactive astrocytes in the brain; an organizer of cerebral ischemia. Neurochem Int 119:107-114) that has not been even mentioned in the present work. 

Considering the experience of Authors (the group of Dr. Koizumi) in the field, I recommend the revision of the manuscript according to the following specific points. 

1)     Please revise and update the references. Nucleotide receptor field and particularly P2X7R have attracted much attention and new review articles are continuously appearing including the recent findings, i.e. Ref 17 could be replaced by Sperlágh and Illes 2014 P2X7 receptor: an emerging target in central nervous system diseases. Trends Pharmacol Sci 35(10):537-47. Some references can be eliminated (Refs 24 and 25) and other ones should be included as concerning the neuroprotective roles of P2X7R in cerebellar granule neurons and intracellular cascades mediating these actions). 

2)     The structure of the paper should be modified. I strongly recommend including a section reviewing the implication of neuronal and glial nucleotide receptors in brain ischemia/hypoxia previous to focus on P2X7R. There are many studies describing the neuroprotector role displayed by P2Y receptors in neurons and glial cells against ischemic events, oxidative stress and glutamate excitotoxicity that have not been mentioned in present work. 

3)     Please revise and reorganize the sections 2, 3 y 4 according to the new one included. As mentioned above neuroprotective roles of P2X7R in granule neurons and intracellular cascades activates must be also included and discussed. 

4)     Please revise and complete Table I. Include neuroprotective role and implication in neurological disorders like depression. 

Author Response

Responses to Reviewer #1

We thank the Reviewer for their time and comments regarding our manuscript and for the detailed suggestions. As described below, we have added additional information to address the points raised by the Reviewer. Please see the responses to the comments below for more details.

Comment:

The paper according to the title appeared to be interesting. However, after its reading and revision, I consider that it requires an extensive revision and reorganization. Some statements/sentences along the manuscript do not agree with data reported in the literature. For example, in the Introduction section “Although glial cells have crucial roles in regulating central nervous system (CNS) injury and recovery, their contribution to cerebral ischemic tolerance remains largely unknown”. This is not true. There is a great body of evidences showing the involvement of glial cells in brain ischemia and hypoxic events as well as the implication of different nucleotide receptors. Moreover, authors have recently reported a review article summarizing the roles of astrocytes in brain ischemia (Koizumi S, Hirayama Y and Morizawa YM, 2018, New roles of reactive astrocytes in the brain; an organizer of cerebral ischemia. Neurochem Int 119:107-114) that has not been even mentioned in the present work. 

Considering the experience of Authors (the group of Dr. Koizumi) in the field, I recommend the revision of the manuscript according to the following specific points. 

Response:

We thank Reviewer #1 for this comment, and we agree with the comment. We have revised our manuscript according to the reviewer’s advice in the revised version of the manuscript (line 42-47).

Comment:

1)     Please revise and update the references. Nucleotide receptor field and particularly P2X7R have attracted much attention and new review articles are continuously appearing including the recent findings, i.e. Ref 17 could be replaced by Sperlágh and Illes 2014 P2X7 receptor: an emerging target in central nervous system diseases. Trends Pharmacol Sci 35(10):537-47. Some references can be eliminated (Refs 24 and 25) and other ones should be included as concerning the neuroprotective roles of P2X7R in cerebellar granule neurons and intracellular cascades mediating these actions). 

Response:

We thank Reviewer #1 for this comment. We have updated the references and added the information on the neuroprotective roles of P2X7R (line 66, 73, 102-108).

Comment:

2)     The structure of the paper should be modified. I strongly recommend including a section reviewing the implication of neuronal and glial nucleotide receptors in brain ischemia/hypoxia previous to focus on P2X7R. There are many studies describing the neuroprotector role displayed by P2Y receptors in neurons and glial cells against ischemic events, oxidative stress and glutamate excitotoxicity that have not been mentioned in present work. 

Response:

We thank Reviewer #1 for this comment, and we agree with the comment. We have added the information on the role of other nucleotide receptors in CNS diseases (line 54-59).

Comment:

3)     Please revise and reorganize the sections 2, 3 y 4 according to the new one included. As mentioned above neuroprotective roles of P2X7R in granule neurons and intracellular cascades activates must be also included and discussed. 

Response:

As mentioned above, we have added the information on the neuroprotective roles of P2X7R in CNS diseases (line 102-108).

Comment:

4)     Please revise and complete Table I. Include neuroprotective role and implication in neurological disorders like depression. 

Response:

In the revised version of Table 1, we have added the information on the neuroprotective roles of P2X7R.

Reviewer 2 Report

The article entitled “P2X7 receptors in astrocytes; as a switch for ischemic tolerance” by Hirayama et al focuses the attention on the role of astrocytic P2X7R in ischemic tolerance. This is certainly an interesting field of investigation as P2X7 is an ATP-gated ion channel expressed by several cells and involved in a lot of mechanisms.

The review is quite comprehensive and exhaustive but I think some additional effort is needed to ameliorate the work.

There are some repetitions, it would be better to use different terms or synonyms. For example, line 23-24 researchers…research or line 176-177 such as…such as.

Line 138-141: this paragraph is not very clear; could the authors explain it better?

I think that the title of this review can be improved like this: P2X7 receptors in astrocytes: a switch for ischemic tolerance.

I suggest the authors update the bibliography, there are very few recent papers.

Author Response

Responses to Reviewer #2

We thank the Reviewer for their time and comments regarding our manuscript and for the detailed suggestions. As described below, we have added additional information to address the points raised by the Reviewer. Please see the responses to the comments below for more details.

Comment:

The article entitled “P2X7 receptors in astrocytes; as a switch for ischemic tolerance” by Hirayama et al focuses the attention on the role of astrocytic P2X7R in ischemic tolerance. This is certainly an interesting field of investigation as P2X7 is an ATP-gated ion channel expressed by several cells and involved in a lot of mechanisms.

The review is quite comprehensive and exhaustive but I think some additional effort is needed to ameliorate the work.

There are some repetitions, it would be better to use different terms or synonyms. For example, line 23-24 researchers…research or line 176-177 such as…such as.

Response:

We thank Reviewer #2 for this comment. We have revised these wording (line 25-26, 204). Also, we have improved the menuscipt complehensively.

Comment:

Line 138-141: this paragraph is not very clear; could the authors explain it better?

Response:

We have revised these sentences to make them much clearer (line 150-155).

Comment:

I think that the title of this review can be improved like this: P2X7 receptors in astrocytes: a switch for ischemic tolerance.

Response:

We thank Reviewer #2 for this comment, and we agree with the comment. We have changed the title according to the reviewer’s suggestion.

Comment:

I suggest the authors update the bibliography, there are very few recent papers.

Response:

As per our response to Reviewer #1, we have updated the references (line 66, 73, 102-108).

Round 2

Reviewer 1 Report

After the revision of the new version of the paper, I have found that authors have not made some of the suggested changes especially those related with neuroprotective role of P2X7R in cerebellar granule neurons.

In the new version, authors have included the work reported by Masuch et al (ref 43) describing the neuroprotective role of P2X7R in microglia (line 102-105). However, there is no mention about the neuroprotective role of P2X7 receptors in rat cerebellar granule neurons, which has been advised. What are the reasons? Do authors have any conflict interest with these findings?

Please, include some comment about the papers reported in granule neurons, at least the paper describing neuroprotection against glutamate excitotoxicity in line 105 (Ortega et al 2011, ERK1/2 activation is involved in the neuroprotective action of P2Y13 and P2X7 receptors against glutamate excitotoxicity in cerebellar granule neurons, Neuropharmacology 61(8):1210-21). Ortega et al. (2010) also reported the neuroprotection displayed by P2X7 and NMDA receptors in granule neurons. Cell Mol Life Sci 67(10):1723-33. Without this comment, I do not recommend the acceptation of the manuscript.

Author Response

Comment:

After the revision of the new version of the paper, I have found that authors have not made some of the suggested changes especially those related with neuroprotective role of P2X7R in cerebellar granule neurons.

In the new version, authors have included the work reported by Masuch et al (ref 43) describing the neuroprotective role of P2X7R in microglia (line 102-105). However, there is no mention about the neuroprotective role of P2X7 receptors in rat cerebellar granule neurons, which has been advised. What are the reasons? Do authors have any conflict interest with these findings?

Please, include some comment about the papers reported in granule neurons, at least the paper describing neuroprotection against glutamate excitotoxicity in line 105 (Ortega et al 2011, ERK1/2 activation is involved in the neuroprotective action of P2Y13 and P2X7 receptors against glutamate excitotoxicity in cerebellar granule neurons, Neuropharmacology 61(8):1210-21). Ortega et al. (2010) also reported the neuroprotection displayed by P2X7 and NMDA receptors in granule neurons. Cell Mol Life Sci 67(10):1723-33. Without this comment, I do not recommend the acceptation of the manuscript.

Response:

We thank Reviewer #1 for this comment. We deeoly apologize for the omission of the articles and description you suggested in the previous version of revision. We have newly added the following sentence in the re-revised version of the manuscript.

Line 104-

In cerebellar granule neuron culture, Ortega et al. showed that glutamate-induced cell death is prevented by P2X7R agonist BzATP [44]. They also showed that BzATP elicites neuroprotection of granule neurons via a phosphorylation of GSK3-mediated mechanism(s) [45].